# A Combined Adjuvant TF–Al Consisting of TFPR1 and Aluminum Hydroxide Augments Strong Humoral and Cellular Immune Responses in Both C57BL/6 and BALB/c Mice

**DOI:** 10.3390/vaccines9121408

**Published:** 2021-11-29

**Authors:** Qiao Li, Zhihua Liu, Yi Liu, Chen Liang, Jiayi Shu, Xia Jin, Chuanyou Li, Zhihua Kou

**Affiliations:** 1Department of Bacteriology and Immunology, Beijing Chest Hospital, Capital Medical University, Beijing Tuberculosis and Thoracic Tumor Research Institute, Beijing 101149, China; liqiao@mail.ccmu.edu.cn (Q.L.); capitalliuyi@mail.ccmu.edu.cn (Y.L.); liangchen@mail.ccmu.edu.cn (C.L.); 2Shanghai Public Health Clinical Center, Fudan University, Shanghai 201508, China; Liuzhihua@shphc.org.cn (Z.L.); shujiayi@shphc.org.cn (J.S.)

**Keywords:** adjuvant, TFPR1, combined adjuvant, inbred mouse strains, TLR2

## Abstract

TFPR1 is a novel adjuvant for protein and peptide antigens, which has been demonstrated in BALB/c mice in our previous studies; however, its adjuvanticity in mice with different genetic backgrounds remains unknown, and its adjuvanticity needs to be improved to fit the requirements for various vaccines. In this study, we first compared the adjuvanticity of TFPR1 in two commonly used inbred mouse strains, BALB/c and C57BL/6 mice, in vitro and in vivo, and demonstrated that TFPR1 activated TLR2 to exert its immune activity in vivo. Next, to prove the feasibility of TFPR1 acting as a major component of combined adjuvants, we prepared a combined adjuvant, TF–Al, by formulating TFPR1 and alum at a certain ratio and compared its adjuvanticity with that of TFPR1 and alum alone using OVA and recombinant HBsAg as model antigens in both BALB/c and C57BL/6 mice. Results showed that TFPR1 acts as an effective vaccine adjuvant in both BALB/c mice and C57BL/6 mice, and further demonstrated the role of TLR2 in the adjuvanticity of TFPR1 in vivo. In addition, we obtained a novel combined adjuvant, TF–Al, based on TFPR1, which can augment antibody and cellular immune responses in mice with different genetic backgrounds, suggesting its promise for vaccine development in the future.

## 1. Introduction

Adjuvants are formulated as an important component of vaccines that aim to enhance immune responses and/or change the type of immune responses [1,2,3,4]. Aluminum salts or hydroxide (alum) have been safely and widely applied in human vaccines for nearly one hundred years, since its adjuvant effect was discovered in the 1920s [5,6,7]. After about 70 years, MF59, an oil-in-water emulsion was proved to act as an effective adjuvant for the influenza vaccine [8,9,10]. Since then, more and more immunostimulants have been discovered to have adjuvanticity due to their capacity to stimulate and modulate immune responses, such as toll-like receptor 4 (TLR4) ligand MPLA (3′-O-deacylated monophosphoryl lipid A) and toll-like receptor 9 (TLR9) ligand CpG-ODN, etc. [11,12,13,14]. According to the requirements for various diseases, adjuvants with different design strategies have been developed, and the use of combined adjuvants is the most common and effective method; for example, AS04 (alum plus MPLA) has been approved for the hepatitis B virus vaccine (Fendrix) and human papillomavirus vaccine (Cervarix) [15,16,17,18,19], and AS03 and CpG+Alum showed a good adjuvant effect for the SARS-CoV2 recombinant protein vaccine in a phase I clinical trial [20].

Previously, the PR-1 domain of Triflin (TFPR1) has been demonstrated to be a potent adjuvant for protein and peptide antigens; it can augment Th1-biased antibody- and cell-mediated immune responses by simply mixing with protein (OVA and recombinant HBsAg) or peptide antigens (HIV-1 pep5 envelope). Furthermore, TFPR1 acts as an immune regulator and a novel adjuvant by activating murine immune cells, such as dendritic cells, B cells and macrophages, primarily by stimulating toll-like receptor 2 (TLR2) [21,22]. The adjuvanticity of TFPR1 was evaluated in BALB/c mice in our previous experiments. BALB/c mice and C57BL/6 mice are the most commonly used inbred mouse strains in the laboratory, because they are homozygous and genetically stable and have clear background information [23,24]. However, there are several differences between the two kinds of mouse strains in immunology and when it comes to applications. For example, they have different gene sequences at the H2 site of the classImajor histocompatibility complex (MHC) gene locus: C57BL/6 mice is H2b but BALB/c mice is H2d. Macrophages, DCs and NK cells from C57BL/6 mice produced higher levels of TNF-α, IL-12 and IFN-γ than those from BALB/c mice after exogenous stimulation, which may affect the development of Th1 and Th2 adaptive immunity [25,26,27]. In addition, the different sequences tend to induce a Th1-associated immune response in C57BL/6 mice and a Th2-associated immune response in BALB/c mice [28,29]. BALB/c mice are commonly used in immunological research (being widely used in the development of monoclonal antibodies), as well as research into inflammation and autoimmunity, while C57BL/6 mice are widely used in the study of gene modifications, transgenic research and oncology. Overall, the adjuvanticity of TFPR1 in mice with different genetic backgrounds remains unknown, and its adjuvanticity needs to be improved to fit the requirements for various vaccines.

In the current study, we first compared adjuvanticity of TFPR1 between two inbred mouse strains, BALB/c and C57BL/6 mice, in vitro and in vivo, and demonstrated that TFPR1 activated TLR2 to exert its immune activity in vivo. Then, we evaluated the effectiveness of TFPR1 as a major component of a combined adjuvant, TF–Al, both in BALB/c mice and in C57BL/6 mice, and found that the combined adjuvant TF–Al augmented specific humoral and cellular immune responses for protein antigens (OVA and HBsAg) in mice with different genetic backgrounds. Our results further confirmed that TFPR1 acts as a novel adjuvant primarily by activating TLR2 in mice and proved the priority of a novel combined adjuvant consisting of TFPR1 and alum.

## 2. Materials and Methods

### 2.1. Mice

Female BALB/c mice and C57BL/6 mice were purchased from Beijing HFK Bioscience Co., Ltd. (6–8 weeks old, Beijing, China). TLR2 gene knock-out homozygous mice (tlr2-/- mouse; exon 3 of the TLR2 gene was knocked out using CRISPR/Cas9 technology) were purchased from the Shanghai Model Organisms Center, Inc. (Shanghai, China).

### 2.2. Mouse Immunization and Sample Collection

The combined adjuvant TF–Al was prepared by adsorbing TFPR1 to alum (Alhydrogel^®^ adjuvant 2%, Catalog code: vac-alu-250, InvivoGen) as follows: one part of alum was added to four parts of TFPR1 solution (250 µg/mL in PBS) and mixed well by pipetting up and down for 5 min. Before immunization, TF–Al was mixed thoroughly with an equal volume of antigens in PBS at least for 5 min. The final volume ratio of alum: TFPR1: antigen is 1:4:5, containing 10 µg of TFPR1 in 100 µL per dose.

The adjuvant properties of TF–Al were evaluated in BALB/c or C57BL/6 mice using two antigens, OVA (10 µg per dose, InvivoGen, San Diego, CA, USA), a commonly used model antigen for adjuvant evaluation, and recombinant HBsAg (1 µg per dose, Bersee, Beijing, China). HBsAg is an important glycoprotein on the surface of HBV, which is widely used for the HBV vaccine [30]. The mice (n = 5 per group) were immunized intramuscularly with TF–Al plus antigen, TFPR1 (10 µg) plus antigen, alum (InvivoGen, San Diego, CA, USA) plus antigen, and immunized twice (HBsAg) or thrice (OVA) with a 3-week interval between injections. Antigen alone and endotoxin-free PBS (GIBCO, Grand Island, NY, USA) were used as controls. Sera were collected before immunization and on the second week after each immunization to detect specific antibodies by ELISA. Splenocytes were isolated at week 3 post final immunization to examine the cellular immune response in ELISPOT assays.

### 2.3. ELISA

Specific anti-antigen antibodies IgG, IgG1 and IgG2a were detected in serum by ELISA, as described previously [31]. Plates were coated with OVA (final concentration, 1 µg/mL) or HBsAg (final concentration, 0.5 µg/mL) to capture specific antibodies. The antibody titer was defined as the reciprocal of the largest dilution of serum for which the OD_450_ value was greater than OD_450negative serum_+2 SD.

### 2.4. ELISPOT Assay for IFN-γ and IL-4

Mouse splenocytes were prepared under sterile conditions, as described previously [21]. Splenocytes were cultured in complete RPMI 1640 (GIBCO, Grand Island, NY, USA) supplemented with 10% FBS, 1% penicillin/streptomycin, and 1% L-glutamine (GIBCO, Grand Island, NY, USA) at 37 °C in an atmosphere of 5% CO_2_ (unless stated otherwise).

IFN-γ- and IL-4-producing splenocytes from mice were detected using mouse IFN-γ or IL-4 ELISPOT assays (BD Biosciences, San Diego, CA, USA), according to the manufacturer’s recommendations. Splenocytes (5 × 10^5^ cells/well in 96-well plates) were incubated in triplicate with or without HBsAg or OVA (final concentration, 10 µg/mL) for 48 h. The spots were counted by an ELISPOT Reader (CTL, Cleveland, OH, USA), and the data were expressed as the number of spot-forming units (SFU) per million cells.

### 2.5. Stimulation of Mouse Splenocytes with TFPR1 In Vitro

Purified TFPR1 (final concentration, 10 µg/mL) was added to splenocytes (5 × 10^5^ cells/well in 96-well plates) either isolated from healthy BALB/c mice or C57BL/6 mice, and Pam3CSK4 (1 µg/mL) was used as a positive control. Then, cells were incubated, supernatants were harvested after 24 h, and the levels of IFN-γ, IL-6, IL-8 and IL-10 were measured using specific ELISA kits (Neobioscience, Shenzhen, China), according to the manufacturer’s instructions. Cytokine concentration was read from the standard curves and expressed as pg/mL.

### 2.6. Detection of Cytokine Levels In Vivo Immunized with TFPR1

BALB/c mice, C57BL/6 mice or TLR2-KO mice were immunized intramuscularly with TFPR1 (10 µg), Pam3CSK4 (10 µg) or PBS. Sera were collected after 3 h or 24 h; IFN-γ, IL-6, IL-8 and IL-10 were measured using specific ELISA kits according to the protocol, as mentioned above.

### 2.7. Statistical Analysis

Statistical analysis was performed using GraphPad Prism, version 5.0 (GraphPad Software, San Diego, CA, USA). The means or geometric means from multiple groups were compared using one-way analysis of variance (ANOVA). The error bars of all the figures were based on standard deviation (SD). A *p* value of less than 0.05 was considered statistically significant.

## 3. Results

### 3.1. TFPR1 Acts as an Effective Adjuvant for the Model Antigen OVA in C57BL/6 Mice

In our previous studies, it was found that the recombinant protein TFPR1 could augment effective immune responses in BALB/c mice [21]. Here, we evaluated the adjuvanticity of TFPR1 in C57BL/6 mice. As in the BALB/c mice, TFPR1 augmented both humoral and intracellular immune responses with a Th1-bised type. C57BL/6 mice immunized with OVA plus TFPR1 had higher titers of anti-OVA IgG antibody (*p* < 0.001 vs. OVA) (Figure 1A), Th2-associated subclass IgG1 antibody (*p* < 0.001 vs. OVA) (Figure 1B) and Th1-associated subclass IgG2a antibody (*p* < 0.001 vs. OVA) (Figure 1C). Unlike the BALB/c mice, C57BL/6 mice immunized with OVA plus TFPR1 generated significantly high numbers of IFN-γ-secreting cells (*p* < 0.01 vs. OVA) (Figure 1D). By contrast, alum induced a Th2-biased antibody responses (Figure 1C) and greater amounts of IL-4 (Figure 1E). These results showed that TFPR1 has similar adjuvanticity for OVA in C57BL/6 mice and BALB/c mice [31], though it induces higher levels of cellular immune responses.

### 3.2. TFPR1 Activates Splenocytes from BALB/c Mice or C57BL/6 Mice to Produce Cytokines with Different Profiles

We next compared cytokine profile differences between C57BL/6 and BALB/c mice in vitro by stimulating mouse splenocytes with TFPR1, and a TLR2/1 agonist Pam3CSK4 was used as a positive control. As shown in Figure 2, TFPR1 stimulated splenocytes from C57BL/6 to secrete significantly higher levels of IFN-γ (*p* < 0.001) (Figure 2A) and IL-6 (*p* < 0.01) (Figure 2B) but reduced production of IL-8 (*p* < 0.001) (Figure 2C) and IL-10 (*p* < 0.05) (Figure 2D) compared to splenocytes from BALB/c mice; the positive control, Pam3CSK4, activated splenocytes to generate similar but different cytokine patterns from TFPR1 in two strains of mice; IFN-γ was at lower concentrations in the supernatant from C57BL/6 splenocytes than it was in the supernatant from BALB/c spleen cells (Figure 2A). Meanwhile, cytokine profiles were observed to be different between the two kinds of mice in vivo; higher levels of IFN-γ (*p* < 0.01) (Appendix A), IL-6 (*p* < 0.001) (Appendix A), IL-8 (*p* < 0.01) (Appendix A) and IL-10 (*p* < 0.001) (Appendix A) were measured in the sera of C57BL/6 mice than those of BALB/c mice. Taken together, TFPR1 is able to activate immune cells to produce proinflammatory and regulatory cytokines in both C57BL/6 and BALB/c mice, but with different patterns, which may explain the difference in the immune responses elicited by TFPR1.

### 3.3. TFPR1 Activates TLR2-KO Mice to Generate Lower Levels of Cytokines than Those in C57BL/6 Mice

Our previous studies indicated that TFPR1 activated murine immune cells primarily by stimulating TLR2 using splenocytes from TLR2-KO mice in vitro [21]. In this study, we further verified the role of TLR2 in the adjuvanticity of TFPR1 in vivo. TLR2-KO mice and wild-type C57BL/6 mice were intramuscularly administered with TFPR1 or Pam3CSK4. Pam3CSK4 promoted the secretion of IFN-γ (*p* < 0.01) (Figure 3A), IL-6 (*p* < 0.001) (Figure 3B), IL-8 (*p* < 0.01) (Figure 3C) and IL-10 (*p* < 0.001) (Figure 3D) in C57BL/6 mice, but not in TLR2-KO mice. Unlike Pam3CSK4, TFPR1 still stimulated TLR2-KO mice to secret most cytokines except IL-6 (Figure 3B) but with significantly lower levels (IFN-γ, IL-8, and IL-10). Taken together, these results further proved that TFPR1 acts as an efficient adjuvant by activating immune cells primarily, but partially through TLR2.

### 3.4. TF–Al Augments Anti-OVA-Specific Humoral and Cellular Immune Responses in C57BL/6 Mice

It has been demonstrated that TFPR1 enhanced the immunogenicity of protein and peptide antigens and induced Th1-biased antibody- and cell-mediated immune responses [21,31]. In this study, we first looked to determine whether TF-Al, which is a combination of TFPR1 and adjuvant alum, augments different immune responses compared to TFPR1 or alum alone for OVA in C57BL/6 mice. Mice immunized with OVA plus TF–Al had higher titers of anti-OVA IgG antibody (*p* < 0.001 vs. TFPR1, *p* < 0.01 vs. alum) (Figure 4A), a subclass IgG1 antibody (*p* < 0.01 vs. TFPR1, *p* < 0.05 vs. alum) (Figure 4B) and IgG2a antibody (*p* < 0.05 vs. alum) (Figure 4C). Furthermore, mice immunized with OVA plus TF–Al generated significantly higher numbers of IFN-γ-secreting cells (*p* < 0.05) (Figure 4D) and low numbers of IL-4-secreting cells (*p* < 0.05) (Figure 4E) than those immunized with OVA plus alum. Taken together, these results demonstrated that TF–Al induced OVA-specific Th1-biased antibody responses and cellular immune responses represented by IFN-γ in C57BL/6 mice compared to alum and that it induced stronger antibody responses than TFPR1 alone.

### 3.5. TF–Al Acts as an Effective Adjuvant for the Model Antigen OVA in BALB/c Mice

Subsequently, we explored the adjuvanticity of TF–Al in BALB/c mice with different genetic backgrounds. Mice immunized with OVA plus TF–Al had higher titers of anti-OVA IgG antibody (*p* < 0.05) (Figure 5A), IgG1 antibody (*p* < 0.01 vs. alum) (Figure 5B) and IgG2a antibody (*p* < 0.001 vs. alum) (Figure 5C) than those immunized with alum plus OVA. They also had higher levels of antibody IgG (*p* < 0.01) and IgG1 (*p* < 0.001) but similar levels of antibody IgG2a compared with those immunized with TFPR1 plus OVA. Results suggested that the combined adjuvant TF–Al can initiate a Th1-biased response which is different from the Th2-skewed response induced by alum and induce a higher level of antibody responses than TFPR1 alone in both BALB/c mice and C57BL/6 mice. TF–Al can also induce a significant augmentation of cellular immune responses in C57BL/6 mice.

### 3.6. TF–Al Augments Anti-HBsAg-Specific Humoral and Cellular Immune Responses in BALB/c Mice

We next examined its adjuvant effect for the recombinant HBsAg. BALB/c mice were intramuscularly immunized with HBsAg plus the combined adjuvant TF-Al, and humoral and cellular immune responses were measured on the second week post immunization. TF–Al significantly augmented antibody responses, including HBsAg-specific IgG (Figure 6A), IgG1 (Figure 6B) and IgG2a (Figure 6C) antibodies, and cellular immune responses, including IFN-γ (Figure 6D) but not IL-4 (Figure 6E), elicited by HBsAg. These results suggested that TF–Al induced HBsAg-specific Th1-biased antibody responses and cellular immune responses dominated by IFN-γ, which is similar to TFPR1 alone, but induced stronger antibody responses. The data presented above clearly indicated that TF–Al is an effective combined adjuvant, which is better than TFPR1 or alum alone.

## 4. Discussion

Vaccines are effective tools to prevent and control infectious diseases in both animals and humans [32,33,34,35], and it is most important that suitable adjuvants are chosen in their design. An ideal adjuvant often has multiple properties: safety, low cost, induction of the expression of related cytokines, an ability to activate humoral and cell-mediated immune responses [36,37,38,39,40]. Combined adjuvants are mainly composed of two parts: immunostimulants and delivery systems. Immunostimulants can activate the antigen-presenting cells (APCs) and promote the secretion of a series of functional cytokines, such as toll-like receptor (TLR) agonists [41,42]. Suitable delivery systems can present antigens to APCs and provide a depot at the injection site for their slow release and continuous stimulation, as with, e.g., alum, emulsions and liposomes [43,44]. The combination of these two features generates synergistic effects, leading to the improvement of vaccine efficacy. Several combined adjuvants have been applied to different vaccines against various infectious diseases and cancers [45,46,47]. For example, AS04, which consists of MPLA and alum, is already used as part of the registered HBV and HPV vaccines, as mentioned above, and AS01, which is composed of MPLA and QS-21, formulated in liposomes, has been applied successfully in malaria (Mosquirix) and herpes zoster (Shingrix) vaccines too [48,49,50,51]. In addition to these licensed vaccines, a protein subunit vaccine candidate for SARS-CoV2 (SCB-2019) contains a stabilized trimeric form of spike protein combined with either AS03 or CpG+Alum, and both adjuvanted vaccine formulations elicited robust humoral and cellular immune responses against SARS-CoV2, with high viral neutralizing activity and being well tolerated in a phase I clinical trial [20]. In this study, we evaluate a combined adjuvant based on our novel adjuvant TFPR1 [31] and alum in BALB/c mice and C57BL/6 mice.

The recombinant protein TFPR1 has been proved to be an effective immunostimulant for protein and peptide antigens in BALB/c mice [21]. Using antibody blocking assays and stimulatory experiments in vitro, we found that TFPR1 activated murine immune cells partially through TLR2. Herein, we further proved that TFPR1 is a TLR2 agonist in vivo and demonstrated its adjuvanticity in C57BL/6 mice, suggesting that TFPR1 has the potential to be used as an important component of combined adjuvants. On the basis of our previous findings, we designed a combined adjuvant, named TF-Al, which is a combination of TFPR1 and alum, and evaluated its adjuvanticity using OVA and recombinant HBsAg as model antigens in vivo. We first determined whether TF–Al could enhance the immunogenicity of OVA in C57BL/6 mice [52,53,54]. Immunization of mice with TF–Al plus OVA augmented both Th1- and Th2-associated antibody and cellular immune responses. Then, we tested whether TF–Al has the same functions in BALB/c mice. The results demonstrated that TF–Al effectively increased concentrations of OVA- and HBsAg-specific antibodies and conferred a similar immunity to TFPR1. Importantly, TF–Al showed stronger adjuvanticity than TFPR1 and alum alone. As expected, the combination of TFPR1 and alum augments specific humoral and cellular immune responses through synergistic effects. Thus, we successfully obtained an effective and promising combined adjuvant, TF-Al.

Importantly, both TFPR1 and TF–Al showed good adjuvanticity in two widely used inbred mice, C57BL/6 and BALB/c mice, although there were some differences in immune responses between the two inbred mice which have different genetic backgrounds. Our results indicated that TFPR1 induced Th1-biased Th1/Th2 mixed antibody responses in C57BL/6 mice and BALB/c mice, and TFPR1 augmented stronger cellular immune responses, represented by IFN-γ, in C57BL/6 mice. Differences in immune responses in BALB/c and C57BL/6 mice may be explained by the different cytokine patterns in the two inbred mice in vivo and in vitro. C57BL/6 mice stimulated with TFPR1 secreted more Th1-type cytokines (IFN-γ and IL-6) and lower levels of IL-8 and IL-10 than BALB/c mice.

Furthermore, using TLR2-KO mice with the C57BL/6 genetic background, we further verified the role of TLR2 in adjuvant activity of TFPR1. Knocking out TLR2 leads to significantly less production of cytokines in mice with TFPR1 stimulation but not the complete suppression of cytokine production, as shown in mice stimulated with a TLR2/1 agonist, Pam3CSK4, indicating that TFPR1 might activate other pathways to play its role as an adjuvant.

In conclusion, we not only verified the primary and partial role of TLR2 in the TFPR1 adjuvant effect, but also successfully obtained a combined adjuvant TF–Al which can enhance strong humoral and cellular immune responses and alter immune types in two inbred mouse strains. Our research provides another adjuvant option for use in human vaccines in the future. One limitation of this study is that only one immunization route (i.m.) was used. We therefore aim to evaluate the adjuvanticity of TF–Al using different routes (e.g., a subcutaneous route) in the next study.

## Figures and Tables

**Figure 1 vaccines-09-01408-f001:**
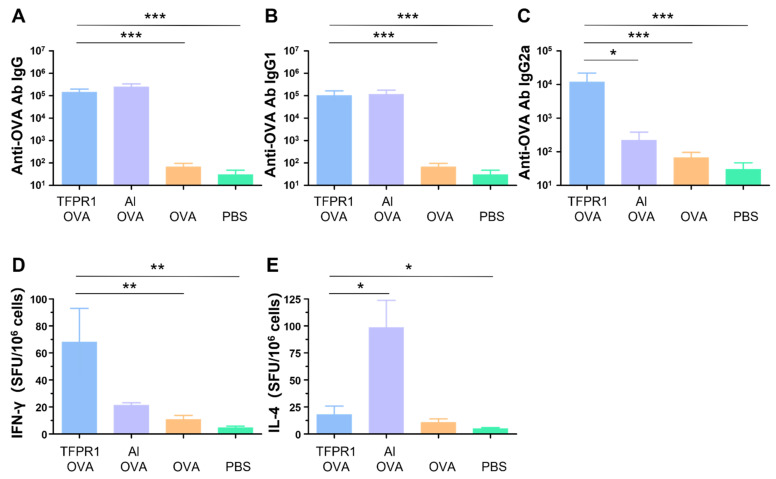
Anti-OVA antibodies and cellular immune responses in C57BL/6 mice immunized with TFPR1 and OVA. C57BL/6 mice were immunized thrice with TFPR1 and OVA, alum and OVA, OVA alone or PBS intramuscularly. On the second week after the final immunization, anti-OVA antibodies were detected by ELISA and cellular immune responses were measured by ELISPOT. (**A**) Anti-OVA antibody IgG. (**B**) Anti-OVA antibody IgG1. (**C**) Anti-OVA antibody IgG2a. (**D**) Specific IFN-γ. (**E**) Specific IL-4. Note: * stands for *p* < 0.05, ** *p* < 0.01, and *** *p* < 0.001.

**Figure 2 vaccines-09-01408-f002:**
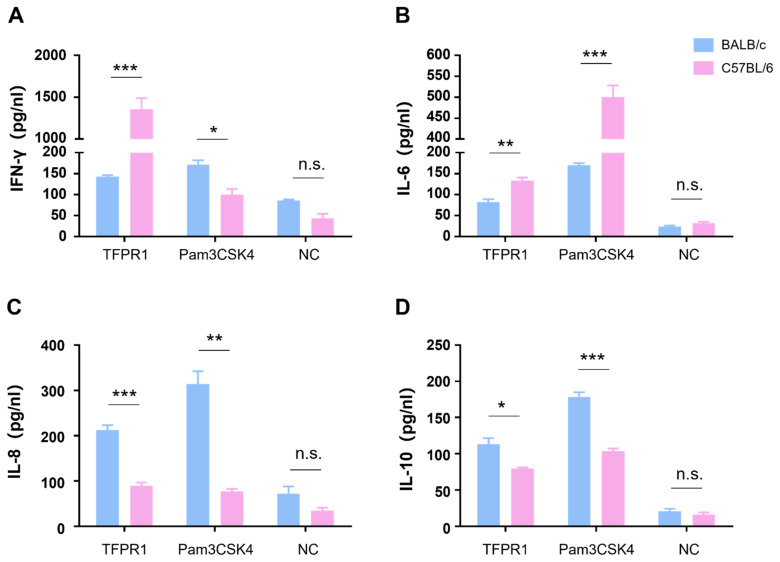
Detection of cytokines secreted by splenocytes from BALB/c and C57BL/6 mice. Splenocytes either from BALB/c or C57BL/6 mice were stimulated with TFPR1 or Pam3CSK4 for 24 h, and cytokines in culture supernatants were measured using ELISA kits. (**A**) IFN-γ. (**B**) IL-6. (**C**) IL-8. (**D**) IL-10. Note: * stands for *p* < 0.05, ** *p* < 0.01, and *** *p* < 0.001. n.s.: no significance.

**Figure 3 vaccines-09-01408-f003:**
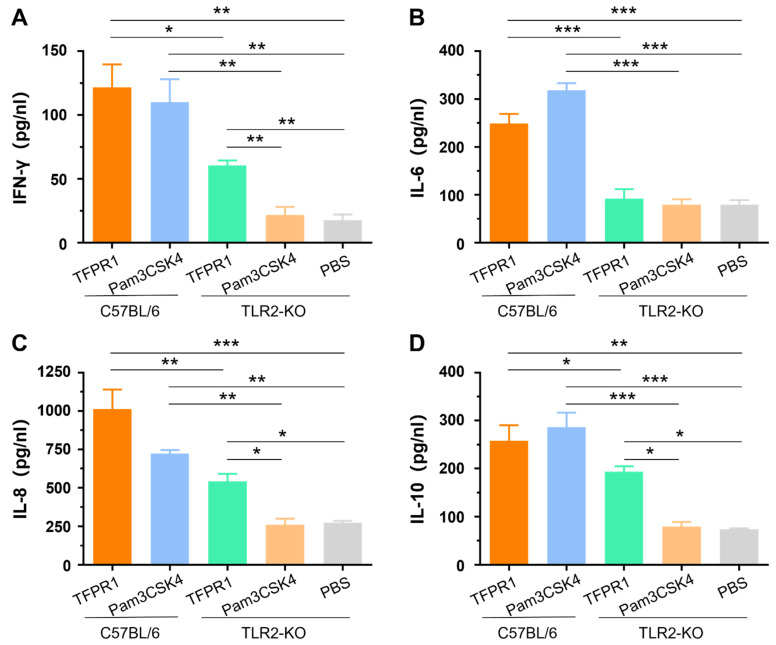
The role of TLR2 in activating cytokines using TLR2-KO (tlr2-/-) mice. C57BL/6 mice and TLR2-KO mice were immunized with TFPR1 or Pam3CSK4 intramuscularly, serum was collected after 24 h and then the levels of cytokines were measured using specific ELISA kits. (**A**) IFN-γ. (**B**) IL-6. (**C**) IL-8. (**D**) IL-10. Note: * stands for *p* < 0.05, ** *p* < 0.01, and *** *p* < 0.001.

**Figure 4 vaccines-09-01408-f004:**
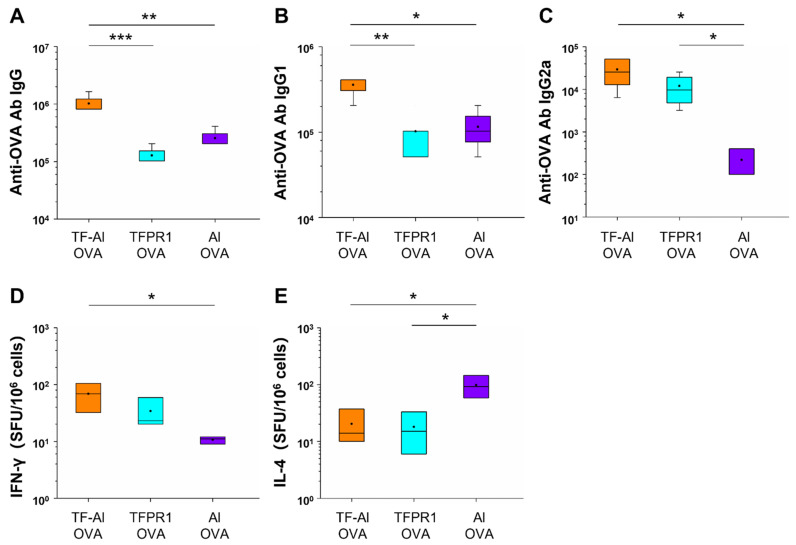
Detection of anti-OVA antibodies and cellular immune responses in C57BL/6 mice immunized with TF–Al and OVA. C57BL/6 mice were immunized thrice with TF–Al plus OVA, TFPR1 plus OVA or alum plus OVA intramuscularly. On the second week after the final immunization, anti-OVA antibodies were detected by ELISA and cellular immune responses by ELISPOT. (**A**) Anti-OVA antibody IgG. (**B**) Anti-OVA antibody IgG1. (**C**) Anti-OVA antibody IgG2a. (**D**) Specific IFN-γ. (**E**) Specific IL-4. Note: * stands for *p* < 0.05, ** *p* < 0.01.

**Figure 5 vaccines-09-01408-f005:**
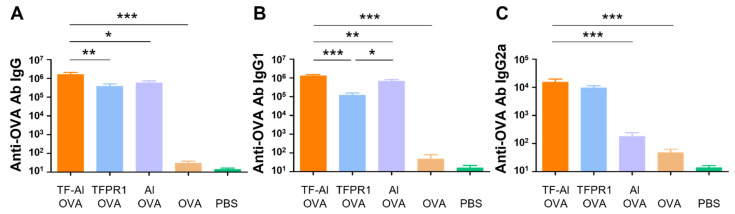
Anti-OVA antibodies in BALB/c mice immunized with TF–Al and OVA. BALB/c mice were immunized thrice with TF–Al and OVA, TFPR1 and OVA, alum and OVA, OVA alone or PBS intramuscularly. Anti-OVA antibodies were detected on the second week after the final immunization by ELISA. (**A**) Anti-OVA antibody IgG. (**B**) Anti-OVA antibody IgG1. (**C**) Anti-OVA antibody IgG2a. Note: * stands for *p* < 0.05, ** *p* < 0.01, and *** *p* < 0.001.

**Figure 6 vaccines-09-01408-f006:**
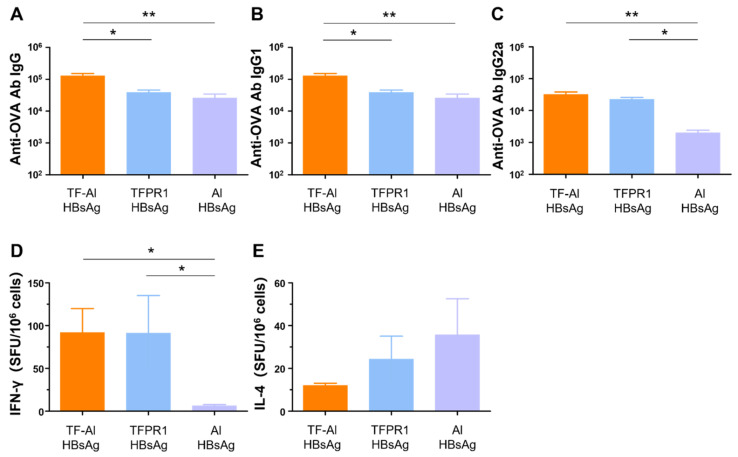
Anti-HBsAg antibodies and cellular immune responses in BALB/c mice immunized with TF–Al plus HBsAg. BALB/c mice were inoculated twice with TF–Al plus HBsAg, TFPR1 plus HBsAg or alum plus HBsAg intramuscularly. On the second week after the final immunization, anti-HBsAg antibodies were detected by ELISA, and cellular immune responses were by ELISPOT. (**A**) Anti-HBsAg antibody IgG. (**B**) Anti-HBsAg antibody IgG1. (**C**) Anti-HBsAg antibody IgG2a. (**D**) Specific IFN-γ. (**E**) Specific IL-4. Note: * stands for *p* < 0.05, ** *p* < 0.01, and *** *p* < 0.001.

## Data Availability

Not applicable.

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
