# Peer review of "A Combined Adjuvant TF–Al Consisting of TFPR1 and Aluminum Hydroxide Augments Strong Humoral and Cellular Immune Responses in Both C57BL/6 and BALB/c Mice"

_vaccines, 2021, doi:10.3390/vaccines9121408_

Round 1

Reviewer 1 Report

In this manuscript, TFPR1 was proven as an effective vaccine adjuvant in both BALB/c mice and C57BL/6 mice and the study further proved the role of TLR2 in playing immune activity in vivo. A novel combined adjuvant EzAd01 based on TFPR1 was obtained, which can augment antibody and cellular immune responses in mice with different genetic backgrounds, suggesting its promising potent for vaccine development in the future.

I recommend publishing the study after answering the following:

1- Did the authors consider the subcutaneous route instead of the intramuscular for immunization ?

2- Please mention the nature of the error bars of all the figures in the manuscript.

3- In Figure 4: Some colors are v pale, so please use other colors and enhance the contrast of the figure.

4- The English language and phrasing needs revision throughout the text. For example in the abstract: "its promising potent" should be changed to "its promising potency".

Reviewer 2 Report

This manuscript by Li et al., presents the evaluation of a combined adjuvant, TFPR1 and alum, in both BALB/c and C57BL/6 mice.  While this study is interesting, the manuscript can be improved further since it seemed to lack scientific rationales and details in the current form.  

-The authors well explained their results and figures, but please elaborate on scientific rationales and significance. For examples, the authors could explain why OVA and recombinant HBsAG were selected as model antigens and what physiological responses are expected for readers who may not be familiar with them.  

-Could the authors provide details on the rationale, why the adjuvanticity in mice with different genetic backgrounds is important? 

-Also could the authors include how this study and its outcome can be clinically relevant and why it is significant?

-Explaining differences in immune responses for BALB/c and C57BL/6 mice in the introduction section will be helpful, rather than in the discussion.

-Lines 289-292,  TFPR-ta doesn't seem to fit in this discussion. -What is TLR2's role?  -Define acronyms when used the first time.

-The title includes EzAd01, but why is TF-Al used throughout the manuscript?  -And what are the ratio of TFPR1 and alum used for studies?
